# Blood pressure and bladder cancer risk in men by use of survival analysis and in interaction with *NAT2* genotype, and by Mendelian randomization analysis

**Stanley Teleka**[1]*, **George Hindy**[2,3], **Isabel Drake**[4], **Alaitz Poveda**[4], **Olle Melander**[4], **Fredrik Liedberg**[5,6], **Marju Orho-Melander**[4], **Tanja Stocks**[1]

1 Department of Clinical Sciences in Lund, Lund University, Lund, Sweden, 2 Department of Population Medicine, College of Medicine Qatar University, Doha, Qatar, 3 Broad Institute, Cambridge, Massachusetts, United States of America, 4 Department of Clinical Sciences in Malmö, Lund University, Lund, Sweden, 5 Division of Urological Research, Institution of Translational Medicine, Lund University, Malmö, Sweden, 6 Department of Urology, Skåne University Hospital, Skåne, Sweden

* stanley.teleka@med.lu.se

**Data Availability Statement:** For the UK-biobank, data are held by the UK Biobank (http://www.ukbiobank.ac.uk/), and can be accessed using the

## Abstract

The association between blood pressure (BP) and bladder cancer (BC) risk remains unclear with confounding by smoking being of particular concern. We investigated the association between BP and BC risk among men using conventional survival-analysis, and by Mendelian Randomization (MR) analysis in an attempt to disconnect the association from smoking. We additionally investigated the interaction between BP and N-acetyltransferase-2 (*NAT2*) rs1495741, an established BC genetic risk variant, in the association. Populations consisting of 188,167 men with 502 incident BC's in the UK-biobank and 27,107 men with 928 incident BC's in two Swedish cohorts were used for the analysis. We found a positive association between systolic BP and BC risk in Cox-regression survival analysis in the Swedish cohorts, (hazard ratio [HR] per standard deviation [SD]: 1.14 [95% confidence interval 1.05–1.22]) and MR analysis (odds ratio per SD: 2-stage least-square regression, 7.70 [1.92–30.9]; inverse-variance weighted estimate, 3.43 [1.12–10.5]), and no associations in the UK-biobank (HR systolic BP: 0.93 [0.85–1.02]; MR OR: 1.24 [0.35–4.40] and 1.37 [0.43–4.37], respectively). BP levels were positively associated with muscle-invasive BC (MIBC) (HRs: systolic BP, 1.32 [1.09–1.59]; diastolic BP, 1.27 [1.04–1.55]), but not with non-muscle invasive BC, which could be analyzed in the Swedish cohorts only. There was no interaction between BP and *NAT2* in relation to BC on the additive or multiplicative scale. These results suggest that BP might be related to BC, more particularly MIBC. There was no evidence to support interaction between BP and *NAT2* in relation to BC in our study.

## Introduction

Elevated blood pressure (BP) is an established risk factor for cardiovascular diseases [1]. Owing to shared risk factors and pathophysiological pathways, several hypotheses have been

reference number 'UK Biobank Main Application 42410'. For the Swedish cohort, due to ethical and legal restrictions related to the Swedish Biobanks in Medical Care Act (2002:297) and the Personal Data Act (1998:204), data are available upon request from the data access group of Malmo Diet and Cancer study and the Malmö Preventive Program by contacting Anders Dahlin (anders. dahlin@med.lu.se).

**Funding:** Funding for this specific study was received from the Crafoord Foundation (no. 20170534) by TS. The funders had no role in study design, data collection and analysis, decision to publish, or preparation of the manuscript.

**Competing interests:** The authors have declared that no competing interests exist.

formed linking BP with cancer [2]. Regarding bladder cancer (BC), studies in human experimental biology have speculated that the angiotensin-renin system, a physiologic pathway responsible for the regulation of BP, may be involved in BC carcinogenesis [3, 4]. We recently reported epidemiologic support of this hypothesis in a large prospective study that showed a positive association between BP and BC risk, but only among men [5]. Other observational studies of BP and BC risk have shown conflicting results, with some studies showing a positive association [5–8], and others showing no association [2, 9–11], altogether resulting in null results in a meta-analysis that included studies predating our previous study [9]. However, most included studies were hampered by limited sample size and a combined analysis of men and women, who could have different risk profiles as indicated by the results in our study [5] and by the substantially higher BC incidence among men than among women [5, 12]. Further, factors interacting with BP in relation to BC might also have caused inconsistent results between studies. N-acetyltransferase 2 (*NAT2*) is a gene that codes for a carcinogen-metabolizing enzyme. The polymorphism that phenotypically expresses "slow acetylation" has been associated with BC, and the interaction between *NAT2* and smoking in relation to BC is well documented [13, 14]. It has been stated that if two exposures are associated with a common outcome, then they must interact either on a multiplicative or additive scale [15]. A potential interaction between BP and *NAT2* in relation to BC has not been investigated.

Mendelian randomization (MR) analysis is a methodological approach that makes use of genetic variants as an instrumental variable (IV) to, under certain assumptions, study the causal association between an exposure of interest and an outcome [16, 17]. A valid IV must fulfill three key assumptions: it must 1) be associated with the exposure of interest, 2) associate with the outcome exclusively through the exposure of interest, and 3) not be associated with confounders in the exposure-outcome association. When these assumptions are met, MR analysis overcomes the major limitations such as residual and unknown confounding, reverse causation and measurement error that are inherent to other observational studies [16, 17]. In relation to BP and BC risk, residual confounding by tobacco smoking, the strongest risk factor for BC [18], is of particular concern. To our knowledge, there are no MR studies on BP and BC risk.

The aim of the study was to investigate the association between BP and BC risk using both conventional survival analysis and MR analysis, and to study the interaction between BP and *NAT2* (rs1495741) in the association. Due to limited statistical power among women in the interaction analysis and MR analysis, which were the added novelty of this study compared to prior studies, we undertook the main investigation among men only.

## Materials and methods

### Study populations

The study included participants from two cohorts in the city of Malmö, in the southernmost part of Sweden, the Malmö Diet and Cancer Study (MDCS) and the Malmö Preventive Project (MPP), and the UK-biobank from the United Kingdom. The MDCS is a population-based cohort of 30,447 participants aged between 45 and 73 years, who underwent a health examination in 1991–96. The MPP is also population-based and included 33,346 men and women who had a health examination in 1974–1992. Detailed descriptions of the Malmö cohorts are published elsewhere [19, 20]. The UK-biobank is a publicly available research resource in the form of a population-based cohort of men and women aged between 40 and 69 years. The project recruited 502,627 individuals nationally between 2006 and 2011. A detailed description of the cohort is published elsewhere [21].

## Ethical considerations

This study was performed in accordance with the Declaration of Helsinki. Participants provided a written consent at baseline physical examination to have their data used for research. The ethics committee at Lund University approved the study of the MDCS and the MPP (Dnr 2014/830). The UK-biobank's research ethics committee and Human Tissue Authority Research Bank approved this study (application number 42410) [22].

## BP assessment

In the MDCS and MPP, BP was measured twice in a recumbent position after a rest of 5 (MDCS) or 10 (MPP) minutes using a standard mercury sphygmomanometer on the right arm, the average of these two values was recorded as the actual levels of BP. In the UK-biobank, two BP readings were taken with the participant seated, with 1-minute interval between readings. An Omron 7015 IT electronic BP monitor (OMRON Healthcare, Europe B.V. Kruisweg 577 2132 NA Hoofddorp) was used to take the readings.

## Follow-up and outcome assessment

In the MDCS and MPP, participants were linked to the national cancer register, cause of death register and the total population register, through their civil registration number, unique to all inhabitants of Sweden. These registers identified cancer diagnoses, death and emigration, respectively. Follow-up for these linkages ended on 31 December 2016. In the UK-biobank, linkages to the UK national cancer registers and cause of death registers were used to identify cancer diagnoses and cause of death, respectively. Information on emigration was obtained from several sources, including the National Health Service. BC was defined according to the ninth edition of the International Classification of Diseases (ICD-9) code 188 [0–9], and ICD-10 code C67 [0–9], including carcinoma in situ (D090). TNM-classification based on histology, palpation and radiology reported to the Swedish National Register of Urinary BC (SNRUBC) was available in the Swedish cohorts. The SNRUBC became nationwide in 1997, and since then has covered on average 97% of BC cases as compared to the Swedish Cancer Register [23]. BC tumors are divided into two groups, based on depth of invasion: 1) Non-muscle invasive BC: Ta, Tis and T1, and 2) Muscle invasive BC: T2, T3, and T4. Death was defined as BC (ICD-10, C67) if recorded as the primary cause of death in the national cause of death registers.

## Genotyping

In the MDCS cohort, a MALDI-TOF mass spectrometer (Sequenom MassArray, Sequenom, San Diego, CA, USA) was used to genotype DNA samples using Sequenom reagents and protocols. In the case where a candidate SNP failed the genotyping, a "proxy SNP" was used in its place. Proxy SNPs were identified using SNAP version 2.2.2 when commercial primers were not available. SNPs that failed Sequenom genotyping were alternatively genotyped individually using TaqMan, KASPar allelic discrimination on an ABI 7900HT (Applied Biosystems, Life Technologies, Carlsbad, CA, USA), per manufacturer's instructions. In the MPP, blood samples were taken, on average, 25 years after study baseline, and was thus excluded from the MR analysis to avoid collider bias [24, 25]. In the UK-biobank, Affymetrix (ThermoFisher Scientifics) performed genotype calling on two closely related, but custom-designed arrays. Approximately 50,000 participants were ran on UK BiLEVE Axiom array and the remaining 450,000 were ran on UK-Biobank Axiom array. A detailed description of the genotype process and internal quality control is described elsewhere [21].

## Mendelian randomization analysis-assumptions

In Mendelian randomization analysis, three key assumptions regarding the IV must be fulfilled. Firstly, it must be associated with the exposure of interest. Secondly, it must be associated with the outcome exclusively through the exposure of interest, and thirdly, it must not be associated with confounders in the exposure-outcome association. In this study, we addressed the first assumption by only using genetic variants that have shown an association with BP in genome-wide association studies (GWAS). Pleiotropy occurs when the IV affects the outcome through a different biological pathway from the exposure of interest. Inclusion of pleiotropic SNPs violates the second assumption, which may lead to biased causal estimates [26]. We investigated for pleiotropy using MR-Egger and MR-PRESSO. Lastly, we addressed the third assumption by investigating the association between the IV and confounders in the BP-BC association, due to the importance of smoking as a confounder, we additionally investigated for potential overlap of genetic variants between the IV and smoking using the most recent GWAS on smoking [27].

## Selection of genetic variants for the systolic BP (SBP) and diastolic BP (DBP) genotype risk scores

Single nucleotide polymorphisms (SNPs) are the most common form of genetic variation in humans. We used a genetic score of BP SNPs as IV in our MR analysis. In the MDCS cohort, a SBP instrument of 29 SNPs with established associations from two large consortia (International consortium of BP genome-wide association studies [ICBP] and the CHARGE consortium) of European ancestry was created [28–30]. Previous MR studies on BP in the MDCS based their IVs on these 29 SNPs [31–33] and a detailed description of the genotype process is reported therein. In the UK-biobank, we created a SBP instrument of 47 SNPs and a DBP instrument of 50 SNPs. The SNPs were obtained from the results provided by the ICBP and 14 other consortia. All SNPs were discovered in populations of European ancestry and outside the UK-biobank [28–30], the latter in order to avoid biased causal estimates towards the confounded observational association, due to the overlap that occurs between the sample that was used to discover the SNP, and the sample used in the MR analysis [16]. We initially found 67 SBP SNPs and 71 DBP SNPs that underwent a rigorous selection process to be included in the instruments; the details are documented in Supplementary information (S1 and S2 Files). In brief, we removed SNPs that were highly correlated (linkage disequilibrium [LD] $\geq$ 0.8), had low genotype rate (<95%), had low minor allele frequency ($\leq$1%), or were out of HWE (threshold calculated as 0.05/number of SNPs tested). Where necessary, a suitable proxy SNP (LD$\geq$0.8) was used for candidate SNPs not available in the UK-biobank. LDlink, a web-based interactive tool was used to find suitable proxy SNPs [34, 35]. The quality control was performed on PLINK v1.9 [36]. To avoid false-positive findings and winner's curse, all the included SNPs had been validated through an independent replication process.

## *NAT2* genotype

To investigate *NAT2* in interaction with BP and BC, we use the SNP"rs1495741 (A/G)". *NAT2* was genotyped in the same way as the BP SNPs per cohort. The polymorphism "A/A" represented fast acetylation, "A/G" represented intermediate acetylation and "G/G" represented slow acetylation (risk variant). In the analysis, we combined fast and intermediate acetylators to investigate *NAT2* polymorphism as a dichotomy.

## Selection of study participants

The combination of MDCS and MPP resulted in 50,670 participants from which 27,107 were included in the final analysis (S1 Fig). The causes of exclusion were cohort overlap, female sex and missing data on SBP, DBP and smoking status. The UK-biobank overall contained 502,543 individuals. In order to mitigate the effects of population stratification, 92,909 individuals who were of Non-European ancestry were excluded from this study. This was achieved through a Principal Component Analysis conducted in all 502,543 participants[22] the causes of exclusion were female sex and missing data on SBP, DBP and smoking status, after which 188,167 participants were retained in the study. In our primary analysis, prevalent BC cases at the time of baseline examination were excluded (44 in the Malmö cohort and 514 in the UK-biobank). In an additional MR analysis, we included prevalent BC cases and women, respectively. The exclusion of women in the main analysis was due to very weak statistical power owing to only 182 incident BCs among women in the MDCS and 129 in the UK-biobank. Furthermore, findings from the largest prospective studies indicated no association among women [5, 7]. A description of the baseline characteristics among women is shown in the supplementary material (S1 Table).

## Statistical analysis

In survival analysis of BP level and BC risk, participants were followed from the baseline examination until the date of event, or until censoring due to diagnosis of another cancer, emigration, or death, whichever one occurred first. The analysis of NMIBC and MIBC in the Swedish cohorts started on 1 January 1997, and censored participants before then were excluded. We used Cox proportional hazards regression to calculate hazard ratios (HR) for BC by SBP and DBP standard transformed (z-scores), per 10 mmHg, and in quartiles. Attained age was used as the underlying time variable, and we adjusted for smoking in five categories (never-smoker, ex-smoker, and tertiles of pack-years among current-smokers), BMI (quintiles), age at baseline (categories) and date of birth (categories). Models in the MDCS and MPP were tested for the additional inclusion of anti-hypertensive medication, physical activity and education; however, adding these co-variables to the model did not change the results, so for consistency with analyses in the UK-biobank, these variables were excluded from further analyses. We tested the proportional hazards assumption using Schoenfield residuals, and found that "age at baseline" and "date of birth" violated the PH assumption; however, inclusion of these variables in the stratum did not materially change the results, so the final models were left un-stratified. The Swedish cohorts combined and the UK-biobank were analyzed separately due to markedly different associations between BP and BC risk. In relation to these findings, we also performed an ad hoc Kaplan-Meier analysis to compare BC-specific survival in the two cohorts to detect any major differences in the proportion of MIBC (S2 Fig). With average length of follow-up of 22 years and 5 years in the Swedish cohorts and UK-biobank, respectively, the leading time between measurement of BP and BC diagnosis likely differed between these cohorts. We therefore calculated the average age at diagnosis among BC cases and performed a lag-time analysis to investigate potential reverse causation in the association between SBP and BC.

We used the quantity "relative excess risk of interaction" (RERI) as our measure of additive interaction between BP and *NAT2* in relation to BC risk, which was based on adjusted HRs. It was calculated by RR11—RR10—RR01 + 1, reflecting the individuals in the lower half of BP and fast/intermediate *NAT2* acetylation (1, reference group), upper half of BP and fast/intermediate *NAT2* acetylation (RR10), lower half of BP and slow *NAT2* acetylation (RR01), and upper half of BP and slow *NAT2* acetylation (RR11). Confidence intervals were obtained using the delta method by Hosmer and Lemeshow. In addition, we investigated multiplicative

interaction between BP and *NAT2* in relation to BC risk using the likelihood ratio test. For the interaction tests, BP and *NAT2* were assessed as categorical variables.

MR analysis can be performed in a one-sample setting, or in a two-sample setting. We first employed the one-sample, 2-stage least square (2SLS) method to estimate associations between genetic scores of the BP indices and BC risk. In the first stage, a weighted genetic score was created as follows: each SNP was coded 0, 1, 2 according to the number of BP-increasing alleles, then that value was weighted according to its effect estimate (β-coefficient) obtained from the aforementioned genome-wide association studies (GWAS), then the weighted value of each SNP were summed up (weighted score = $[\beta_1 \times SNP_1 + \beta_2 \times SNP_2 + \ldots \beta_n \times SNP_n]$/number of SNPs). Next, we regressed the weighted genetic score on the z-transformed BP levels (SBP or DBP). The predicted values, corresponding to the predicted z-transformed genetic level of SBP or DBP, were used as IV in MR analyses of BC risk. Additionally, we performed MR in a two-sample setting, with the added advantage of formally testing for pleiotropy. We used the inverse-variance weighted (IVW) estimation to investigate the association between BP and BC using two-sample MR analysis. It is obtained from the linear regression of the genetic associations with BC on the genetic associations with BP indices using inverse variance weights and the intercept restrained to zero in the model. To detect pleiotropy, we performed the MR-Egger test and MR-PRESSO. The MR-Egger estimate is similar to the IVW except that the intercept is left unrestrained. It provides accurate estimates even in the presence of an invalid instrument, but is limited by the InSIDE (Instrumental strength independent of direct effects) assumption and can only detect the direction of pleiotropy (cannot detect presence of pleiotropy in opposing direction) [17]. Pleiotropy is suggested if the Egger intercept is significantly different from zero. MR-PRESSO is a tool designed to evaluate horizontal pleiotropy in a two-sample setting. It has three components and the first component (MR-PRESSO global test) detects horizontal pleiotropy [37]. Additionally, we evaluated the influence of any potentially outlying SNPs in the MR-Egger estimates using a leave-one out analysis. The two-sample analyses were performed using the STATA package "mrrobust" [38] and R packages "TwoSampleMR" and "MR-PRESSO" [37]. We also investigated the associations between the IVs and potential confounders, and between the BP indices and potential confounders, by linear/logistic regression (S2 Table). Some IVs were associated with body mass index (BMI); however, the variance explained for BMI by the BP GSs was only 0.02–0.05%. Furthermore, we searched for other traits associated with the SNPs that may be linked with BC through other biological pathways. These analyses were performed on phenoscanner v2 [39], an online, publicly available database containing results from large-scale genetic associations in humans. In phenoscanner, genetic variants are cross-referenced for associations with a wide-range of other traits. All the statistical analyses were performed in STATA 13, (StataCorp LLC, College Station, TX) and RStudio version 1.1.423.

## Results

There were 27,107 men in the Swedish cohorts and 188,167 men in the UK-biobank. Mean age at baseline was 58 years (SD = 8) amongst men in the UK-biobank and 50 years (SD = 11, Table 1) in the Swedish cohorts. Approximately 12% of men in the UK-biobank were current smokers at baseline, compared to 43% of men in the Swedish cohorts. On average, men in the UK-biobank had a SBP level of 143 mmHg (SD = 19) and a DBP level of 84 mmHg (SD = 11), and the corresponding in the Swedish cohorts were 135 mmHg (SD = 19) and 87 mmHg (SD = 10), respectively. Furthermore, 58% of the men in the UK-biobank had hypertensive BP levels (SBP/DBP ≥140/90) compared to 53% in the Swedish cohorts, and 26% of the men the UK-biobank were obese (BMI ≥30 kg/m$^2$) compared to only 10% in the Swedish cohorts.

**Table 1. Baseline characteristics of the study participants included in the assessment of the risk of bladder cancer in relation to blood pressure.**

| Characteristic | MDCS and MPP (n = 27,107) | UK-biobank (n = 188,167) |
|---|---|---|
| **Baseline year, range** | 1974–1996 | 2006–2010 |
| **Baseline age, years, mean (SD)** | 50.4 (10.7) | 57.7 (8.1) |
| **Category, n (%)** | | |
| <30 | 533 (2.0) | 0 (0.0) |
| 30–44 | 7,168 (26.4) | 17,904 (9.5) |
| 45–59 | 13,273 (49.0) | 81,881 (43.5) |
| ≥60 | 6,133 (22.6) | 88,382 (47.0) |
| **Smoking status, n (%)[*]** | | |
| Never smoker | 8,024 (30.6) | 91,735 (48.9) |
| Ex-smoker | 7,010 (26.8) | 73,528 (39.2) |
| Current smoker | 11,172 (42.6) | 22,230 (11.9) |
| **Pack years among current smokers, n (%)[*]** | | |
| <10 | 1,611 (18.8) | 2,305 (13.5) |
| 10–19.9 | 925 (10.8) | 3,312 (19.4) |
| ≥20 | 6,043 (70.4) | 11,470 (67.1) |
| **Blood pressure, mm Hg, mean (SD)** | | |
| Systolic blood pressure | 134.9 (19.1) | 143.3 (18.5) |
| Diastolic blood pressure | 86.7 (9.9) | 84.2 (10.6) |
| **Category, systolic/diastolic, n (%)** | | |
| <140/90 mm Hg | 12,678 (46.8) | 78,832 (41.9) |
| 140/90-159/99 mm Hg | 9,304 (34.3) | 70,676 (37.6) |
| ≥160/100 mm Hg | 5,125 (18.9) | 38,659 (20.5) |
| **BMI, kg/m$^2$, mean (SD)[†]** | 25.4 (3.6) | 27.9 (4.2) |
| <18.5 | 280 (1.0) | 422 (0.2) |
| 18.5–24.9 | 12,891 (47.6) | 46,418 (24.8) |
| 25–29.9 | 11,286 (41.6) | 92,943 (49.6) |
| ≥30 | 2,634 (9.8) | 47,758 (25.6) |
| **Mean follow-up time, years (SD)** | 22.2 (11.5) | 4.8 (3.9) |
| **Follow-up time, n (%)** | | |
| <5 | 2,192 (8.1) | 53,878 (28.6) |
| 5–9 | 2,224 (8.2) | 134,289 (71.4) |
| 10–14 | 2,668 (9.8) | 0 (0.0) |
| ≥15 | 20,023 (73.9) | 0 (0.0) |

[*] Smoking status was missing for 674 (0.4%) men in the UK-biobank and for 901 (3.3%) men in the MDCS and MPP combined. Includes accumulated pack-years among current smokers,
Excluding 2 593 (9.6%) and 5 143 (2.7%) current smokers with missing pack-years data in the MPP and MDC combined and UK-biobank respectively.
[†] BMI data were missing for 626 men in the UK-biobank and 16 men in MDCS and MPP combined.
Abbreviations: MDCS, Malmö Diet and Cancer Study; MPP, Malmö Preventive Program; BMI, body mass index.

During a mean follow-up time of five years (SD = 4) in the UK-biobank, 502 incident BCs occurred, and during a mean follow-up time of 22 years (SD = 12) in the Swedish cohorts, 928 incident BCs occurred.

Table 2 shows the HRs for BC overall and separately for NMIBC and MIBC (in the Swedish cohorts only) by continuous z-scores, per 10 mmHg and in quartiles of SBP and DBP. SBP,

**Table 2. Hazard ratio (95% confidence interval)[*] of bladder cancer outcomes by levels of systolic and diastolic blood pressure among men.**

| | | MDCS & MPP (N = 27,107) | | | UK-biobank (N = 188,167) |
|---|---|---|---|---|---|
| | | Muscle-invasive bladder cancer | Non-muscle invasive bladder cancer | Bladder cancer incidence | Bladder cancer incidence |
| Exposure | Exposure level | (N cases = 105) [†] | (N cases = 425) [†] | (N cases = 928) | (N cases = 498) |
| **SBP, mm Hg** | Per SD | **1.32 (1.09–1.59)** | 1.06 (0.96–1.18) | **1.14 (1.05–1.22)** | 0.93 (0.85–1.02) |
| | Per 10mm Hg | **1.14 (1.02–1.27)** | 1.02 (0.96–1.08) | **1.05 (1.01–1.09)** | 0.96 (0.92–1.01) |
| | Quartiles | | | | |
| | Q1 | 1.0 (reference) | 1.0 (reference) | 1.0 (reference) | 1.0 (reference) |
| | Q2 | 1.08 (0.60–1.94) | 1.16 (0.87–1.53) | **1.23 (1.01–1.49)** | 1.04 (0.80–1.35) |
| | Q3 | 1.12 (0.65–1.92) | 1.21 (0.91–1.62) | **1.36 (1.12–1.66)** | 0.94 (0.73–1.22) |
| | Q4 | 1.82 (0.97–3.39) | 1.17 (0.86–1.59) | **1.24 (1.00–1.52)** | 0.86 (0.67–1.13) |
| **DBP, mm Hg** | Per SD | **1.27 (1.04–1.55)** | 0.99 (0.89–1.10) | 1.02 (0.95–1.09) | 0.96 (0.91–1.01) |
| | Per 10mm Hg | **1.25 (1.03–1.53)** | 0.99 (0.89–1.10) | 1.02 (0.95–1.09) | 0.98 (0.90–1.07) |
| | Quartiles | | | | |
| | Q1 | 1.0 (reference) | 1.0 (reference) | 1.0 (reference) | 1.0 (reference) |
| | Q2 | 1.08 (0.60–1.94) | 0.99 (0.74–1.32) | 0.96 (0.78–1.17) | 1.04 (0.82–1.32) |
| | Q3 | 1.12 (0.65–1.92) | 1.16 (0.90–1.49) | 1.16 (0.98–1.38) | 1.09 (0.85–1.39) |
| | Q4 | 1.38 (0.81–2.33) | 0.96 (0.73–1.26) | 0.96 (0.80–1.16) | 0.92 (0.71–1.20) |

[*] Hazard ratios were calculated using Cox proportional hazards regression models with attained age as the underlying time scale, adjusted for smoking (categories), age at baseline (categories), date of birth (categories), and BMI (quintiles).

[†] Data on tumor staging was only available in the MDCS and MPP cohorts, it was obtained from the Swedish National Register of Urinary BC (SNRUBC), which originated in 1997. As a result all tumors that occurred before 1997, which were available for the analysis on total incidence, were not included in the analysis for NMIBC and MIBC.

Abbreviations: MDCS, Malmö diet and cancer study; MPP, Malmö preventive project; SD, standard deviation; SBP, systolic blood pressure; DBP, diastolic blood pressure.

but not DBP, was positively associated with overall incidence of BC in the Swedish cohorts, the HR per SD (95% CI) was 1.14 (1.05–1.22). Furthermore, the association between SBP and BC risk overall, in the Swedish cohorts, was stronger for those in the second, third and fourth quartile compared to those in the first quartile. SBP and DBP were both positively associated with MIBC, the HRs per SD were 1.32 (1.09–1.59) and 1.27 (1.04–1.55), respectively. In the UK-biobank, SBP and DBP were not associated with BC risk.

There was no statistically significant additive interaction between BP and *NAT2* in relation to BC in the UK-biobank and MDCS when using RERI as the measure of interaction (Fig 1). Likewise, there was no statistically significant interaction on a multiplicative scale using the LR test; the p-value was 0.82 in the UK-biobank and 0.67 in the MDCS.

The associations between SBP and DBP with BC risk in the MDCS and UK-biobank, determined by 2SLS regression and IVW estimation, are shown in Fig 2. Genetically predicted elevation in SBP was associated with higher BC risk in the MDCS, the odds ratio (OR) (95%CI) per SD was 7.70 (1.92–30.9) for the 2SLS and 3.43 (1.12–10.5) for IVW. Similar to measured BP levels, there were no associations between genetically predicted SBP and DBP levels and BC risk in the UK-biobank. S3–S5 Figs of MR-Egger estimates for SBP and DBP in relation to BC risk showed that the intercept did not significantly differ from zero in any of the analysis assessing for pleiotropy. This was further supported by no evidence of horizontal pleiotropy and outlying SNPs in the MR-PRESSO and leave-one out analysis respectively (S6–S8 Figs). The MR-PRESSO global test had p-values of 0.65, 0.16 and 0.37 for systolic BP in the MDCS, and systolic and diastolic BP in the UK-biobank, respectively. When including prevalent BC

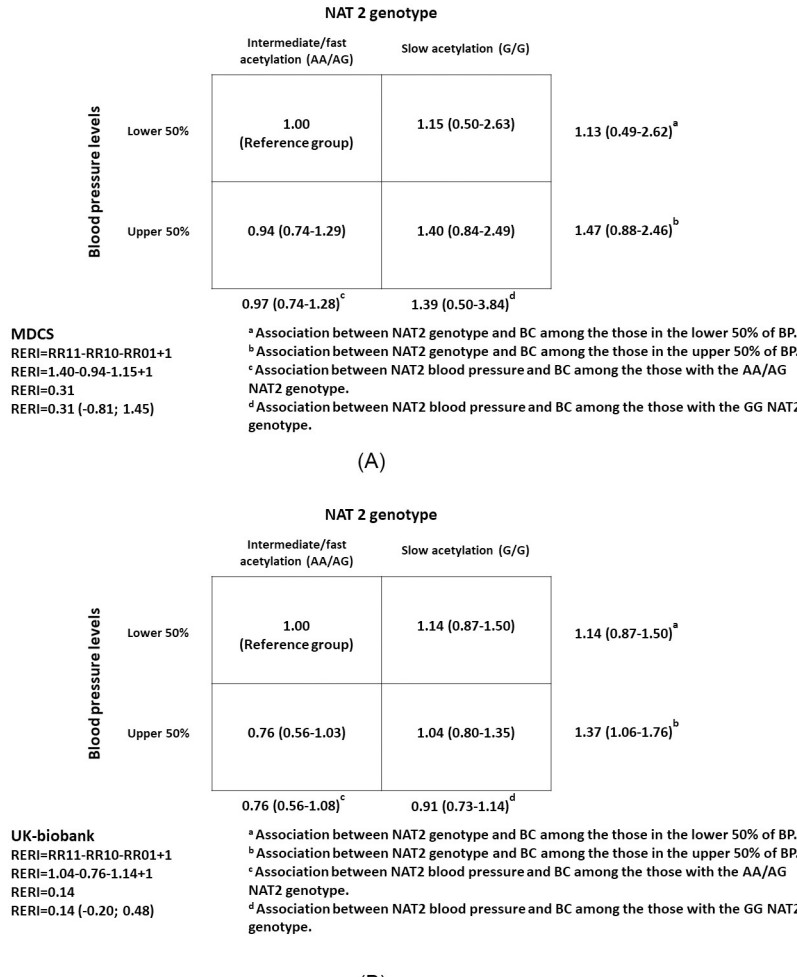

Fig 1. Additive interaction between blood pressure and *NAT2* in relation to bladder cancer risk in the (A) Malmö Diet and Cancer Study (MDCS; $N_{participants}$ = 7 749; $N_{cases}$ = 282) and (B) UK-biobank ($N_{participants}$ = 187 688; $N_{cases}$ = 498).

cases (S3 Table) or women (S4 Table) in the MR analysis, the associations tended to be weaker, although confidence intervals for these results largely overlapped the results for incident BC among men only.

Further investigation followed to understand potential explanations for the different findings between the Swedish cohorts and the UK-biobank. The average age at BC diagnosis was 76 years for the Swedish cohorts and 66 years for the UK-biobank, which could possibly translate to BCs of different tumor characteristics. However, survival curves of incident BC cases in the UK-biobank and the MDCS were similar (p-value for the log-rank test = 0.092) and thus, did not provide a clear explanation for the different findings between the cohorts (S2 Fig). The HRs per SD (95% CI), in the lag-time analysis for SBP and BC risk in the UK-biobank were closer to 1 than the original: 0.97 (0.87–1.09) and 1.00 (0.84–1.19) for 3 and 5 years respectively. Relatively few cases were omitted for the respective analysis in the Swedish cohorts (1.3% for 3 years and 5.6% 5 years), resulting in no material change in HRs.

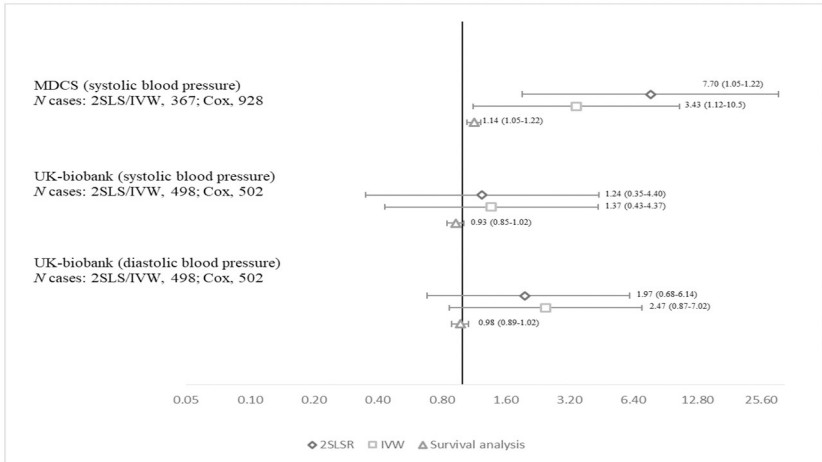

**Fig 2. Relative risk (95% confidence interval) of bladder cancer per standard deviation of systolic and diastolic blood pressure using Mendelian randomization two stage least square regression (2SLSR) regression and inverse variance weighted (IVW) method, and Cox regression**[*]**, in the Malmö Diet and Cancer Study (MDCS) and UK-biobank.** [*]Also includes the Malmö Preventive Project.

## Discussion

In this study, we investigated the association between SBP, DBP and BC risk among men in cohorts in Sweden and the UK-biobank, using conventional and MR analysis. In conventional survival analysis, we found that SBP was positively associated with BC risk overall in the Swedish cohorts, but not in the UK-biobank, and both SBP and DBP were positively associated with MIBC, but not NMIBC, which was investigated in the Swedish cohorts only. We further observed a positive association between SBP and BC risk by MR analysis of men in the MDCS, but not in the UK-biobank. Additionally, we investigated additive and multiplicative interaction between BP and *NAT2* (rs1495741) in relation with BC risk, but did not find any support for such interaction.

The different findings between the cohorts may have several explanations. Participant characteristics of the cohorts differed at large, both with regards to blood pressure levels, BMI and smoking, which altogether might limit the capacity of applying external validity between the two cohorts. Secondly, low participation rate remains a concern in the MDCS, where the participation was 41% [40], but even more so in the UK-biobank, which is known as a very selective population with a participation rate of only 5% [25]. Furthermore, the difference in the average age at diagnosis between the cohorts may suggest a difference in the type of BC occurring. Although survival analysis of BC cases did not indicate major differences in disease aggressiveness between the cohorts, different etiology of BC and the relative importance of risk factors such as BP in younger vs. older age could in part contribute to the different findings. Lastly, the lag-time analysis for 3 and 5 years respectively in the UK-biobank slightly changed HRs, potentially suggesting the influence of reverse causation.

The null association between BP and NMIBC risk and a positive association between BP and MIBC risk in the Swedish cohorts, may suggest that the positive association between BP and BC risk overall in conventional and MR analysis of the Swedish cohorts are largely driven by MIBC tumors. This is further supported by a somewhat weaker association between BP and BC risk in the MR analysis that included prevalent cases, which inherently comprise more indolent BC's. However, the positive association between SBP and BC observed in the MR analysis of the MDCS must be interpreted with caution. On one hand, the result is consistent

with findings from the conventional analysis in this and our previous, larger study [5], and in some other previous observational studies [6–8]. However, the association may also be driven by low study power and pleiotropy. In our study, the MR-Egger test, MR-PRESSO and leave-one out analysis did not indicate pleiotropy, which may be a true reflection, but may also be a result of insufficient statistical power. The use of a stronger IV to predict BP would have been desirable for increased statistical power; however, in the largest BP GWAS to date of 535 loci associated with BP, 325 SNPs were discovered in the UK-biobank. Including SNPs discovered in the UK-biobank would led to sample overlap, which is strongly discouraged in a two-sample analysis due to the high risk of obtaining biased estimates [16, 41]. Furthermore, the 210 remaining SNPs had not been validated, increasing the potential for false positive findings, if included. To validate our findings in the MDCS, further studies are needed based on stronger IVs and a larger number of validated BC cases, ideally separated by muscle invasiveness.

A potential biological mechanism linking BP to BC remains unclear. Studies from experimental biology on human BC cells have suggested that the angiotensin-renin pathway may play a role in BC carcinogenesis [3, 4]. From these studies, it is suggested that the angiotensin-renin pathway might play a role in BC progression, which would support an association between BP and BC driven by MIBC. However, these findings need to be replicated and validated in other population studies.

Despite the use of large cohorts, statistical power was the main weakness of this study. The study was large enough to examine main associations between BP and BC risk in the conventional analysis, but interaction analysis requires more power, which may explain the null interaction observed between BP and *NAT2*. With sufficient power, we expected to see interaction either on an additive or multiplicative scale or both since *NAT2*, through smoking, is a known risk factor for BC, and BP is a potentially independent risk factor for BC. Likewise, limited statistical power in the MR analysis did not allow us to detect effect estimates nearly as low as the estimates in the conventional survival analyses. This would have been counteracted by a meta-analysis of the results from the MDCS and the UK-biobank, which, however, we considered inappropriate given the different findings between the cohorts. The main strengths of the study were the large sample size for the observational analysis, the detailed smoking data, and the investigation of three separate cohorts, which allowed us to investigate the reliability of our results from one cohort on the other.

In conclusion, in this study of BP and BC risk among men, SBP was positively associated with BC risk in both conventional and MR analysis of Swedish cohorts, but not in the UK-biobank. However, the population characteristics differed at large between the cohorts. There was no evidence to support interaction between BP and *NAT2* in relation with BC. The heterogeneous results between the cohorts and low study power in some of the analyses calls for more epidemiological studies in the field.

## Supporting information

**S1 File. Systolic blood pressure SNP selection.**
(XLSX)

**S2 File. Diastolic blood pressure SNP selection.**
(XLSX)

**S1 Fig. Selection of participants in the Malmö Diet and Cancer Study (MDCS), Malmö Preventive Project (MPP) and UK-biobank.**
(TIF)

**S2 Fig. Kaplan Meier curves for bladder cancer-specific survival among incident bladder cancer cases since time of diagnosis by study population.**
(TIF)

**S3 Fig. MR-Egger plots for the (a) inverse variance-weighted (IVW) estimate and (b) MR-Egger estimate for systolic blood pressure, with bladder cancer as the end-point in the Malmö Diet and Cancer Study.**
(TIF)

**S4 Fig. MR-Egger plots for the (a) inverse variance-weighted (IVW) estimate and (b) MR-Egger estimate for systolic blood pressure, with bladder cancer as the end-point in the UK-biobank.**
(TIF)

**S5 Fig. MR-Egger plots for the (a) inverse variance-weighted (IVW) estimate and (b) MR-Egger estimate for diastolic blood pressure, with bladder cancer as the end-point in the UK-biobank.**
(TIF)

**S6 Fig. Leave–one out analysis of 29 systolic blood pressure single nucleotide polymorphisms (SNPs) in the Malmö Diet and Cancer Study.**
(TIF)

**S7 Fig. Leave–one out analysis of 47 systolic blood pressure single nucleotide polymorphisms (SNPs) in the UK–Biobank.**
(TIF)

**S8 Fig. Leave–one out analysis of 50 diastolic blood pressure single nucleotide polymorphisms (SNPs) in the UK–Biobank.**
(TIF)

**S1 Table. Baseline characteristics women in the Swedish cohorts and the UK-biobank.**
(PDF)

**S2 Table. Association between per standard deviation of measured and instrumental variables of systolic and diastolic blood pressure, and potential confounders in the relationship between blood pressure and bladder cancer risk.** Abbreviations: SBP, systolic blood pressure; DBP, diastolic blood pressure; IV, instrumental variable; MDCS, Malmö diet and Cancer Study; OR, odds ratio; BMI, body mass index. [a] For age at baseline, date of birth, BMI, smoking, and education, we used linear regressions to investigate the association with blood pressure indices and their respective genetic scores. For physical activity and antihypertensive medication, we used logistic regression.
(PDF)

**S3 Table. Odds ratio (95% confidence interval) from Mendelian randomization analysis of incident bladder cancer, and incident and prevalent bladder cancers combined, for systolic and diastolic blood pressure in the Malmö Diet and Cancer Study and UK-biobank.**
(PDF)

**S4 Table. Two stage least square regression and inverse variance weighted method for systolic and diastolic blood pressure in relation to bladder cancer incidence for men and women combined in the Malmö Diet and Cancer Study and UK-biobank.** Abbreviations: MDCS, Malmö Diet and Cancer Study; OR, odd ratio; CI, confidence intervals; BP, blood pressure; 2SLS, two-stage least square regression; IVW, inverse-variance weighted. [a] $R^2$ is the

proportion of BP variance that is explained the genetic score.
(PDF)

## Acknowledgments

The authors wish to thank all UK-biobank, MDCS and MPP participants and staff. We thank Anders Dahlin, database manager of the Malmö cohorts, and Joana Howson for technical support of the UK-biobank. This Research has been conducted using the UK-biobank Resource (application number, 42410). The UK-biobank was established by the Welcome Trust Medical Charity, Medical Research, Department of Health, The Scottish Government and Northwest Regional Development Agency.

## Author Contributions

**Conceptualization:** Stanley Teleka, Tanja Stocks.

**Data curation:** Stanley Teleka, Tanja Stocks.

**Formal analysis:** Stanley Teleka.

**Funding acquisition:** Tanja Stocks.

**Investigation:** Stanley Teleka, George Hindy, Isabel Drake, Alaitz Poveda, Olle Melander, Fredrik Liedberg, Marju Orho-Melander, Tanja Stocks.

**Methodology:** Stanley Teleka, George Hindy, Tanja Stocks.

**Writing – original draft:** Stanley Teleka, Tanja Stocks.

**Writing – review & editing:** Stanley Teleka, George Hindy, Isabel Drake, Alaitz Poveda, Olle Melander, Fredrik Liedberg, Marju Orho-Melander, Tanja Stocks.

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
