## [Decision Letter · Decision Letter 0]

25 Aug 2020

PONE-D-20-19309

Blood pressure and bladder cancer risk in men by use of survival analysis and in interaction with NAT2 genotype, and by Mendelian randomization analysis

PLOS ONE

Dear Dr. Teleka,

Thank you for submitting your manuscript to PLOS ONE. After careful consideration, we feel that it has merit but does not fully meet PLOS ONE’s publication criteria as it currently stands. Therefore, we invite you to submit a revised version of the manuscript that addresses the points raised during the review process.

Please respond to the major issues regarding the methods and the results according to the reviewers' comments.

Please ensure that our decision is justified on PLOS ONE’s publication criteria and not, for example, on novelty or perceived impact.

We look forward to receiving your revised manuscript.

Kind regards,

Jeffrey S Chang

Academic Editor

PLOS ONE

Journal Requirements:

2. In the ethics statement in the manuscript and in the online submission form, please provide additional information about the patient records used in your retrospective study, including: a) the date range (month and year) during which patients' medical records were accessed; b) the date range (month and year) during which patients whose medical records were selected for this study sought treatment; and c) the source of the medical records analyzed in this work (e.g. hospital, institution or medical center name). If patients provided informed written consent to have data from their medical records used in research, please include this information.

3.We note that you have indicated that data from this study are available upon request. PLOS only allows data to be available upon request if there are legal or ethical restrictions on sharing data publicly. For information on unacceptable data access restrictions, please see http://journals.plos.org/plosone/s/data-availability#loc-unacceptable-data-access-restrictions.

4. Your ethics statement must appear in the Methods section of your manuscript. If your ethics statement is written in any section besides the Methods, please move it to the Methods section and delete it from any other section. Please also ensure that your ethics statement is included in your manuscript, as the ethics section of your online submission will not be published alongside your manuscript.

Reviewers' comments:

Reviewer's Responses to Questions

**Comments to the Author**

1. Is the manuscript technically sound, and do the data support the conclusions?

Reviewer #1: Partly

Reviewer #2: Yes

2. Has the statistical analysis been performed appropriately and rigorously? 

Reviewer #1: Yes

Reviewer #2: Yes

3. Have the authors made all data underlying the findings in their manuscript fully available?

Reviewer #1: Yes

Reviewer #2: Yes

4. Is the manuscript presented in an intelligible fashion and written in standard English?

Reviewer #1: Yes

Reviewer #2: Yes

5. Review Comments to the Author

Reviewer #1: The manuscript entitled “Blood pressure and bladder cancer risk in men by use of survival analysis and in interaction with NAT2 genotype, and by Mendelian randomization analysis” by Teleka et al. seeks to confirm associations between blood pressure and bladder cancer risk in men, using data from two large prospective cohorts, the Sweden cohort and the UK-biobank. In addition to traditional survival models, the authors constructed genetic scores using significant SNPs based on previous GWAS findings and incorporated in mendelian randomization analysis. These are the strengths of the study. However, the significance of the study is reduced by the following concerns:

1. The study was claimed to address the residual confounding effect of smoking on associations between blood pressure and bladder cancer risk, because smoking is the major risk factor for bladder cancer. However, besides the standard adjustment for smoking status and packyears in the analysis, the study did not take any extra effort to control the residual effect of smoking. Mendelian randomization analysis was primarily applied to genetic scores, not for smoking. The statement in the abstract “we investigated the association between BP and BC risk among men using survival analysis and mendelian randomization analysis in an attempt to disconnect the association from smoking” is misleading.

2. The observed associations differed between Sweden cohort and UK-biobank. It was noted that blood pressure was only a one-time measurement and it was measured in the Sweden cohort in 1974-1996 and in the UK-biobank in 2006-2010. With an average 22 years versus 5 years follow-up time between the two cohorts, the leading times between the blood pressure measurement and bladder cancer diagnosis in the two cohorts were likely differed markedly, which would confound the associations. Discussion on this matter is necessary.

3. It is not clear why an SNP in NAT2 was included in the association analysis of blood pressure and bladder cancer risk. Genetic variants in NAT2 are known to modify the relationships between carcinogen exposure and bladder cancer risk, but not affect blood pressure. Some justifications would help appreciate the significance of the analysis.

Some minor issues:

1. Please define SBP and DBP at the first appearance, page 4, last paragraph.

2. Please check Figure 1A and 1B for RERI calculation. It is identical between the two figures.

Reviewer #2: Overall, this is an interesting manuscript. The authors investigated the association between blood pressure (BP) and bladder cancer (BC) risk through both traditional survival analysis and Mendelian randomized analysis, using two large datasets - UK biobank and the Swedish cohorts. In addition, the authors evaluated the interaction between BP and NAT2 (rs1495741) on the risk of bladder cancer. However, there are a few comments that I believe need additional clarification and consideration.

In lines 49-55, the authors described three assumptions for Mendelian randomization analysis. However, it is unclear whether such instrumental variable assumptions are satisfied in this application. Please provide a clear justification for the assumptions either using biological knowledge (e.g., the Bradford Hill criteria) or statistical testing.

On page 8, why do the authors consider the analysis of pleiotropy? The authors wanted to evaluate the instrumental variables assumptions implicitly, so they tested NAT2 genetic variants with the measured covariates to assess potential pleiotropy? In addition, can the authors clarify whether the causal pathways from NAT2 genetic variants to BC are through the risk factor BP or through pleiotropy (i.e., NAT2 genetic association with other variables is via a different causal pathway and not via BP), or both?

In Table 2, why does the sum of muscle-invasive (N_cases=105) and non-muscle invasive (N_cases=425) bladder cancer cases not equal to the total number of bladder cancer incidence in MDCS and MPP cohorts (N_cases=928)? Due to missing values in tumor staging?

In Figures 1A and 1B, the relative excess risk of interaction (RERI) values do not seem to be consistent with that derived from 2x2 tables for MDCS and UK biobank, respectively. For example, as shown in a 2x2 table of Figure 1A, R10=1, R01=1.15, R11=1.40, and therefore the value of RERI should be calculated by 1.40-1-1.15+1 = 0.25, but not 0.31. Also, for UK biobank in Figure 1B, that does not sound right to me.

For Figure 2, why is the relative risk [or OR] estimate of SBP based on the two stage least square regression (2SLS) method larger than that from the inverse variance weighted (IVW) method using MDCS data in Mendelian randomization analysis? Can the authors provide an explanation?

The caption of Figure 2 stated “relative risk (95% confidence interval)”, but the authors stated “the odds ratio (OR) (95% CI) in lines 269-270. Please fix them (relative risk or OR).

6. PLOS authors have the option to publish the peer review history of their article (what does this mean?). If published, this will include your full peer review and any attached files.

Reviewer #1: No

Reviewer #2: No

---

## [Author Response · Author response to Decision Letter 0]

1 Oct 2020

PONE-D-20-19309

Blood pressure and bladder cancer risk in men by use of survival analysis and in interaction with NAT2 genotype, and by Mendelian randomization analysis

We would like to thank the editor and the reviewers for a thorough and careful reading of this manuscript. We appreciate the constructive feedback, and we have incorporated most of the suggestions into the manuscript. Below, we have answered to each of the points raised. 

Reviewer comments

Reviewer #1

1. The study was claimed to address the residual confounding effect of smoking on associations between blood pressure and bladder cancer risk, because smoking is the major risk factor for bladder cancer. However, besides the standard adjustment for smoking status and packyears in the analysis, the study did not take any extra effort to control the residual effect of smoking. Mendelian randomization analysis was primarily applied to genetic scores, not for smoking. The statement in the abstract “we investigated the association between BP and BC risk among men using survival analysis and mendelian randomization analysis in an attempt to disconnect the association from smoking” is misleading.

We thank the reviewer for this comment, indeed the conventional survival analysis investigating blood pressure in relation to bladder cancer risk may not have fully accounted for residual confounding by smoking. Confounding may never be fully adjusted for in an observational analysis with such a strong confounder as smoking in BC, which is why we conducted the Mendelian randomization analysis. In the Mendelian randomization analysis, we used the blood pressure genetic score as a proxy to actual blood pressure levels in the relationship between blood pressure and bladder cancer. None of the SNPs used in the genetic score were associated with smoking characteristics (such as age at initiation, cigarettes per day and smoking cessation) from the latest GWAS1 or from phenoscanner v2, and the BP genetic score was not associated with smoking (S2_table). We agree that the statement in the abstract appears misleading, and we have changed it to “We investigated the association between BP and BC risk among men using conventional survival-analysis, and by Mendelian Randomization (MR) analysis in an attempt to disconnect the association from smoking…” (Page 1, paragraph 1). The comma before “survival-analysis” indicates that the attempt to disconnect the association from smoking relates to the Mendelian randomization analysis.

2. The observed associations differed between Sweden cohort and UK-biobank. It was noted that blood pressure was only a one-time measurement and it was measured in the Sweden cohort in 1974-1996 and in the UK-biobank in 2006-2010. With an average 22 years versus 5 years follow-up time between the two cohorts, the leading times between the blood pressure measurement and bladder cancer diagnosis in the two cohorts were likely differed markedly, which would confound the associations. Discussion on this matter is necessary. 

We thank the reviewer for raising this relevant point. We agree that the leading times between blood pressure and bladder cancer diagnosis and the difference in length of follow-up between the Swedish cohorts and UK-biobank may explain the difference in the observed associations. Firstly, the shorter leading time in the UK-biobank increased the likelihood of reverse causation. We have now conducted a lag time analysis of 3 and 5 years, and the hazard ratios (per SD) for BC in UK were 0.97 (0.87-1.09) and 1.00 (0.84-1.19) respectively. These results approach those of the Mendelian randomization analysis and Swedish cohorts, which may suggest the influence of reverse causation; however, CIs are wide and overlapping between the various analyses. The lag time analysis for 3 and 5 years in the Swedish cohorts only omitted 1.3% and 5.6%, respectively, consequentially, this did not change the results. Furthermore, the average age at diagnosis among bladder cancer cases differed when comparing the Swedish cohorts and the UK-biobank (76 years in the Swedish cohorts and 66 years in the UK biobank), which might have contributed to the different findings between the cohorts. In relation to the above, we have added statements in the methods, results and discussion:

Methods, page 9, line 202-206:

“With average length of follow-up of 22 years and 5 years in the Swedish cohorts and UK-biobank, respectively, the leading time between measurement of BP and BC diagnosis likely differed between these cohorts. We therefore calculated the average age at diagnosis among BC cases and performed a lag-time analysis to investigate potential reverse causation in the association between SBP and BC.”

Results, page 18, lines 291-293:

“The average age at BC diagnosis was 76 years for the Swedish cohorts and 66 years for the UK-biobank, which could possibly translate to BCs of different tumor characteristics” 

Page 18, lines 295-299 

“The HRs per SD (95% CI), in the lag-time analysis for SBP and BC risk in the UK-biobank were closer to 1 than the original: 0.97 (0.87-1.09) and 1.00 (0.84-1.19) for 3 and 5 years respectively. Relatively few cases were omitted for the respective analysis in the Swedish cohorts (1.3% for 3 years and 5.6% for 5 years), resulting in no material change in HRs.”

Discussion, page 19, lines 315-322:

“Furthermore, the difference in the average age at diagnosis between the cohorts may suggest a difference in the type of BC occurring. Although survival analysis of BC cases did not indicate major differences in disease aggressiveness between the cohorts, different etiology of BC and the relative importance of risk factors such as BP in younger vs. older age could in part contribute to the different findings. Lastly, the lag-time analysis for 3 and 5 years respectively in the UK-biobank slightly changed HRs, potentially suggesting the influence of reverse causation.”

3. It is not clear why an SNP in NAT2 was included in the association analysis of blood pressure and bladder cancer risk. Genetic variants in NAT2 are known to modify the relationships between carcinogen exposure and bladder cancer risk, but not affect blood pressure. Some justifications would help appreciate the significance of the analysis.

Indeed, genetic variants of NAT2 have shown an association with bladder cancer by conferring an additional risk to exposure of carcinogens such as tobacco smoking and have not shown an association with blood pressure. In relation to the concept of scale dependence in interaction, if two risk factors have an effect on an outcome, there has to be interaction either on an additive scale or multiplicative scale, this was described in greater detail from page 72 of Modern Epidemiology (3rd edition)2. We wanted to explore the potential interaction between blood pressure and NAT2 (two independent factors) and bladder cancer (the outcome) both on the additive and multiplicative scale. If blood pressure is indeed causally associated with bladder cancer, we expected to observe interaction on at least one of the two scales. We have added statements in the discussion (page 20, paragraph 3) to clarify this concept (added text in bold)

 “Despite the use of large cohorts, statistical power was the main weakness of this study. The study was large enough to examine main associations between BP and BC risk in the conventional analysis, but interaction analysis requires more power, which may explain the null interaction observed between BP and NAT2. With sufficient power, we expected to see interaction either on an additive or multiplicative scale or both since NAT2, through smoking, is a known risk factor for BC, and BP is a potentially independent risk factor for BC. Likewise, limited statistical power in the MR analysis did not allow us to detect effect estimates nearly as low as the estimates in the conventional survival analyses. This would have been counteracted by a meta-analysis of the results from the MDCS and the UK-biobank, which, however, we considered inappropriate given the different findings between the cohorts.”

Some minor issues:

1. Please define SBP and DBP at the first appearance, page 4, last paragraph.

We have defined SBP and DBP on page 6, line 134-135.

2. Please check Figure 1A and 1B for RERI calculation. It is identical between the two figures.

We have corrected the RERI correction for Figure 1A and 1B.

Reviewer #2

1. In lines 49-55, the authors described three assumptions for Mendelian randomization analysis. However, it is unclear whether such instrumental variable assumptions are satisfied in this application. Please provide a clear justification for the assumptions either using biological knowledge (e.g., the Bradford Hill criteria) or statistical testing.

We have written a paragraph in the methods section describing how we addressed the 3 key assumptions in Mendelian randomization. In brief, to address the first assumption, we only used genetic variants that were associated with blood pressure (achieved genome-wide significance) in genome-wide association studies. We addressed the second assumption making an inquiry for pleiotropy using two methods (MR-Egger and MR-PRESSO). We addressed the third assumption by investigating the association between the instrumental variable (IV) and confounders in the BP-BC association using regression analysis.

Methods, page 6, lines 120-133:

“Mendelian randomization analysis-assumptions

In Mendelian randomization analysis, three key assumptions regarding the IV must be fulfilled. Firstly, it must be associated with the exposure of interest. Secondly, it must be associated with the outcome exclusively through the exposure of interest, and thirdly, it must not be associated with confounders in the exposure-outcome association. In this study, we addressed the first assumption by only using genetic variants that have shown an association with BP in genome-wide association studies (GWAS). Pleiotropy occurs when the IV affects the outcome through a different biological pathway from the exposure of interest. Inclusion of pleiotropic SNPs violates the second assumption, which may lead to biased causal estimates 3. We investigated for pleiotropy using MR-Egger and MR-PRESSO. Lastly, we addressed the third assumption by investigating the association between the IV and confounders in the BP-BC association, due to the importance of smoking as a confounder, we additionally investigated for potential overlap of genetic variants between the IV and smoking using the most recent GWAS on smoking.”

2. On page 8, why do the authors consider the analysis of pleiotropy? The authors wanted to evaluate the instrumental variables assumptions implicitly, so they tested NAT2 genetic variants with the measured covariates to assess potential pleiotropy? In addition, can the authors clarify whether the causal pathways from NAT2 genetic variants to BC are through the risk factor BP or through pleiotropy (i.e., NAT2 genetic association with other variables is via a different causal pathway and not via BP), or both?

We considered analysis for pleiotropy to address the second key assumption for Mendelian randomization analysis (see above). We did not test the NAT2 genetic with measured covariates to assess potential pleiotropy, the motivation for investigating the NAT2 genetic variant was to explore its role as an interaction term (additive and multiplicative interaction) in the blood pressure-bladder cancer association, this is a separate analysis and is not related to the pleiotropic analysis we performed using MR-Egger and MR-PRESSO.

3. In Table 2, why does the sum of muscle-invasive (N_cases=105) and non-muscle invasive (N_cases=425) bladder cancer cases not equal to the total number of bladder cancer incidence in MDCS and MPP cohorts (N_cases=928)? Due to missing values in tumor staging?

Tumor data in this study was obtained from the Swedish National Register of Urinary BC (SNRUBC), which originated in 1997. As a result all tumors that occurred before 1997, which were available for the analysis on total incidence, were not included in the analysis for NMIBC and MIBC.

The following was added as a footnote to Table 2

”… it was obtained from the Swedish National Register of Urinary BC (SNRUBC), which originated in 1997. As a result all tumors that occurred before 1997, which were available for the analysis on total incidence, were not included in the analysis for NMIBC and MIBC.”

4. In Figures 1A and 1B, the relative excess risk of interaction (RERI) values do not seem to be consistent with that derived from 2x2 tables for MDCS and UK biobank, respectively. For example, as shown in a 2x2 table of Figure 1A, R10=1, R01=1.15, R11=1.40, and therefore the value of RERI should be calculated by 1.40-1-1.15+1 = 0.25, but not 0.31. Also, for UK biobank in Figure 1B, that does not sound right to me.

We thank the reviewer for noting these topographical errors and repetition of results, we have now corrected figure 1A and 1B.

5. For Figure 2, why is the relative risk [or OR] estimate of SBP based on the two stage least square regression (2SLS) method larger than that from the inverse variance weighted (IVW) method using MDCS data in Mendelian randomization analysis? Can the authors provide an explanation?

The two stage least square regression (2SLS) has reduced statistical power compared to the Inverse-variance weighted (IVW) method. With an increasing sample size, the results from the 2SLS approximate those of the IVW as demonstrated by the results in the UK-biobank (where the difference in the results for 2SLS and IVW do not differ as much), which has a significantly larger sample size compared to the Malmö diet and cancer study. Furthermore, the confidence interval for the 2SLS and IVW results in the MDCS largely overlap.

6. The caption of Figure 2 stated “relative risk (95% confidence interval)”, but the authors stated “the odds ratio (OR) (95% CI) in lines 269-270. Please fix them (relative risk or OR).

Since both hazard ratios and Odds ratios are displayed on the same graph/plot, we decided to use the inclusive term of “relative risk” as a general term referring to these associations.

References

1. Liu M, Jiang Y, Wedow R, Li Y, Brazel DM, Chen F, Datta G, Davila-Velderrain J, McGuire D, Tian C, Zhan X, Agee M, Alipanahi B, Auton A, Bell RK, Bryc K, Elson SL, Fontanillas P, Furlotte NA, Hinds DA, Hromatka BS, Huber KE, Kleinman A, Litterman NK, McIntyre MH, Mountain JL, Northover CAM, Sathirapongsasuti JF, Sazonova OV, Shelton JF, Shringarpure S, Tian C, Tung JY, Vacic V, Wilson CH, Pitts SJ, Mitchell A, Skogholt AH, Winsvold BS, Sivertsen B, Stordal E, Morken G, Kallestad H, Heuch I, Zwart J-A, Fjukstad KK, Pedersen LM, Gabrielsen ME, Johnsen MB, Skrove M, Indredavik MS, Drange OK, Bjerkeset O, Børte S, Stensland SØ, Choquet H, Docherty AR, Faul JD, Foerster JR, Fritsche LG, Gabrielsen ME, Gordon SD, Haessler J, Hottenga J-J, Huang H, Jang S-K, Jansen PR, Ling Y, Mägi R, Matoba N, McMahon G, Mulas A, Orrù V, Palviainen T, Pandit A, Reginsson GW, Skogholt AH, Smith JA, Taylor AE, Turman C, Willemsen G, Young H, Young KA, Zajac GJM, Zhao W, Zhou W, Bjornsdottir G, Boardman JD, Boehnke M, Boomsma DI, Chen C, Cucca F, Davies GE, Eaton CB, Ehringer MA, Esko T, Fiorillo E, Gillespie NA, Gudbjartsson DF, Haller T, et al. Association studies of up to 1.2 million individuals yield new insights into the genetic etiology of tobacco and alcohol use. Nature Genetics 2019;51(2):237-244.

2. Rothman KJ, Greenland S, Lash TL. Modern Epidemiology Wolters Kluwer Health/Lippincott Williams & Wilkins, 2015.

3. Bowden J, Davey Smith G, Burgess S. Mendelian randomization with invalid instruments: effect estimation and bias detection through Egger regression. Int J Epidemiol 2015;44(2):512-25.

---

## [Decision Letter · Decision Letter 1]

20 Oct 2020

Blood pressure and bladder cancer risk in men by use of survival analysis and in interaction with NAT2 genotype, and by Mendelian randomization analysis

PONE-D-20-19309R1

Dear Dr. Teleka,

We’re pleased to inform you that your manuscript has been judged scientifically suitable for publication and will be formally accepted for publication once it meets all outstanding technical requirements.

Kind regards,

Jeffrey S Chang

Academic Editor

PLOS ONE

Additional Editor Comments (optional):

Reviewers' comments:

Reviewer's Responses to Questions

**Comments to the Author**

1. If the authors have adequately addressed your comments raised in a previous round of review and you feel that this manuscript is now acceptable for publication, you may indicate that here to bypass the “Comments to the Author” section, enter your conflict of interest statement in the “Confidential to Editor” section, and submit your "Accept" recommendation.

Reviewer #1: All comments have been addressed

Reviewer #2: All comments have been addressed

2. Is the manuscript technically sound, and do the data support the conclusions?

Reviewer #1: Yes

Reviewer #2: Yes

3. Has the statistical analysis been performed appropriately and rigorously? 

Reviewer #1: Yes

Reviewer #2: Yes

4. Have the authors made all data underlying the findings in their manuscript fully available?

Reviewer #1: Yes

Reviewer #2: Yes

5. Is the manuscript presented in an intelligible fashion and written in standard English?

Reviewer #1: Yes

Reviewer #2: Yes

6. Review Comments to the Author

Reviewer #1: All my previous concerns were fully addressed and the manuscript was modified accordingly. I have no additional questions.

Reviewer #2: The authors have addressed and clarified the questions I raised in my previous review. I have no further comments to make.

7. PLOS authors have the option to publish the peer review history of their article (what does this mean?). If published, this will include your full peer review and any attached files.

Reviewer #1: No

Reviewer #2: No

---

## [Editor Report · Acceptance letter]

29 Oct 2020

PONE-D-20-19309R1 

Blood pressure and bladder cancer risk in men by use of survival analysis and in interaction with *NAT2* genotype, and by Mendelian randomization analysis 

Dear Dr. Teleka:

I'm pleased to inform you that your manuscript has been deemed suitable for publication in PLOS ONE. Congratulations! Your manuscript is now with our production department. 

Kind regards, 

on behalf of

Dr. Jeffrey S Chang 

Academic Editor

PLOS ONE